

# Aerosol-Cloud Interactions in Marine Low-Clouds in a Warmer Climate

Prasanth Prabhakaran[1,2], Timothy A. Myers[3], Fabian Hoffmann[4], and Graham Feingold[2]

[1]Cooperative Institute for Research In Environmental Sciences (CIRES), University of Colorado, Boulder, CO, USA
[2]Chemical Sciences Laboratory, National Oceanic and Atmospheric Administration, Boulder, CO, USA
[3]Physical Sciences Laboratory, National Oceanic and Atmospheric Administration, Boulder, CO, USA
[4]Ludwig-Maximilans-Universität München, Meteorologisches Institut, Munich, Germany

**Correspondence:** Prasanth Prabhakaran (prasantp@mtu.edu)

**Abstract.** We explore the impact of aerosol perturbation on the stratocumulus-to-cumulus transition (SCT) in a warmer climate in the North-East Pacific region using a Lagrangian large-eddy simulation model coupled to a two-moment, bin-emulating bulk microphysics scheme. We explore two SCT cases with different free-tropospheric (FT) humidities - moist FT and dry FT. For each case, we consider two Shared Socioeconomic Pathways (SSPs), SSP3-7.0 and SSP1-2.6, from the most recent Coupled

Model Intercomparison Project (CMIP6) to determine the extent of warming and changes in aerosol concentration at the end-of-the-century. We find that the cloud radiative effect (CRE) in non-precipitating stratocumulus clouds is more susceptible to climate change than to aerosol. However, after the breakup of the cloud deck, the impact of aerosol tends to dominate. Furthermore, in these low-clouds, aerosol-cloud interactions (Twomey effect and liquid water path adjustments) are to leading order immune to climate change, unless aerosol-induced cloud fraction adjustment is significant. We extend the analysis to

marine cloud brightening and show that its efficacy decreases with warming because of the reduction in cloud fraction. We also explore the impact of climate change and aerosol perturbation on SCT. In the moist FT case, climate change advances the onset of cumulus activity and cloud breakup. However, in the dry FT case, climate change does not affect the onset of cumulus activity but delays cloud breakup. In both cases, aerosol injection delays cloud breakup via precipitation suppression but does not affect cumulus onset unless it is coupled to rain formation.

## 1   Introduction

Marine low-clouds strongly regulate global mean temperature by reflecting a substantial fraction of incoming short-wave (SW) radiation to outer space (Wood, 2012). The climate feedback associated with these clouds is one of the largest uncertainties in Earth's climate projections (Zelinka et al., 2013). The picture becomes more complex once the uncertainties associated with aerosol-cloud interactions are included (Boucher et al., 2013; Bellouin et al., 2020). The key reason for this is the poor

representation of turbulent cloud processes in climate models. In this paper, we explore the impact of climate change on aerosol-cloud interactions in marine low-clouds using Lagrangian (domain moving with mean wind) large-eddy simulations (LESs).



With the advent of high-resolution numerical simulations (LES and cloud-resolving simulations), our understanding of the response of marine low-clouds to climate perturbations has improved substantially (Bretherton et al., 2013; Bretherton and Blossey, 2014). These studies explored the response of marine stratocumulus, cumulus-under-stratocumulus, and shallow cumulus clouds to a warmer climate. They showed that weakening of the subsidence velocity and/or strengthening of the inversion aid in thickening of the stratocumulus deck, a negative feedback with respect to cloud radiative effect (CRE) (Bretherton, 2015). However, weakened radiative cooling and warming result in a strong positive feedback. The net cloud feedback is positive in the stratocumulus and cumulus-under-stratocumulus cases, and weakly positive in the cumulus regime. Their analysis also explored the thermodynamic mechanism behind the thinning and reduction in cloud cover ($f_c$) in marine stratocumulus clouds in a warmer world that results in a positive cloud feedback (entrainment-liquid flux feedback). A similar framework was also used to explore cloud feedback in precipitating and spatially organized marine cumulus clouds (Vogel et al., 2016). Furthermore, several studies have attempted to observationally constrain marine low-cloud feedback. To do so, they quantify the sensitivity of marine low clouds to meteorological perturbations via satellite observations, and then use climate models to estimate meteorological changes with planetary warming. The observation-based findings from these studies are in qualitative agreement with high-resolution models (Cesana and Del Genio, 2021; Ceppi et al., 2024; Myers et al., 2021; Klein et al., 2017, and references therein).

Over the last two decades, several studies have focused on understanding the physical processes associated with aerosol-cloud interactions (ACI) in marine low-clouds. The Twomey effect, an increase in cloud albedo in response to an increase in cloud droplet concentration ($N_d$) while assuming the LWP is constant, and the life-time effect, where the cloud albedo is increased by suppressing precipitation, have been well explored (Twomey, 1974; Albrecht, 1989). LES studies have highlighted that decrease in sedimentation velocity (sedimentation-entrainment feedback) and the increase in evaporation rate (evaporation-entrainment feedback) at the cloud-top in response to increases in $N_d$ result in an increase in the intensity of turbulence at the cloud-top (Wang et al., 2003; Ackerman et al., 2004; Bretherton et al., 2007). This increases the entrainment rate at cloud-top that eventually results in a reduction in LWP. However, these studies do not account for the effects of the diurnal cycle. Using an ensemble of LES simulations, Zhang et al. (2024) argue that the effects of SW absorption may buffer some of the aforementioned cloud adjustments in response to aerosol perturbations. Moreover, Prabhakaran et al. (2024) showed that weak precipitation suppression causes cloud darkening due to enhanced SW absorption. However, not many studies have focused on how ACI would change in a warmer climate, which is the central question addressed here. Changes in boundary layer turbulence as well as changes in baseline cloud properties due to climate change may alter the cloud response to aerosol. The aerosol perturbation we use in this study is a proxy for intermittent emissions from volcanoes, ship emissions, or a deliberate injection of aerosol to enhance the reflectivity of marine low-clouds (marine cloud brightening, MCB). MCB is a climate intervention approach aimed at mitigating some of the worst effects of anthropogenic radiative forcing (Feingold et al., 2024). The insights from this study should also be useful in assessing the efficacy of MCB in a warmer world.

We use the stratocumulus-to-cumulus transition (SCT) as a framework for exploring ACI in a warmer climate. Consequently, we also explore the impact of climate change on SCT as few studies have explored this topic. The stratocumulus-topped boundary layer transitions into a cumulus-topped boundary layer as the air mass advects towards the equator over a continuously





increasing ocean temperature. The associated increase in surface latent heat flux (LHF) and the weakening of subsidence pro-
motes the deepening and decoupling of the cloud-topped boundary layer (Bretherton, 1992; Krueger et al., 1995; Wyant et al.,

1997). Eventually, the formation of overshooting cumulus clouds erode the decoupled stratus layer at the top of the boundary
layer. We refer to this transition as an entrainment-driven transition. Recent studies have shown that precipitation also likely
plays a key role in the breakup of the stratocumulus cloud layer (Yamaguchi et al., 2017; Zhou et al., 2017; Sarkar et al., 2020).
We refer to these transitions as the precipitation-mediated transition. In this study, we explore the impact of climate change and
aerosol perturbation on both precipitation-mediated and entrainment-driven SCTs.

In the next section, we present our methodology for setting up cases in a warmer world. This is followed by the results
from the simulations exploring the impact of climate change and aerosol perturbation on SCT in both precipitation-mediated
and entrainment-driven scenarios. We end with a discussion of our results focusing on the impact of climate change on cloud
properties, SCT, ACI, and MCB, followed by a summary.

## 2  Methodology

The setup of the LES follows the framework presented in Yamaguchi et al. (2017) and Prabhakaran et al. (2024), and thus will
only be briefly discussed. All simulations have a domain size of 48 km in the horizontal directions with a uniform grid size
of 100 m. The domain top is 4.25 km in the vertical with a grid size of 10 m until a height of 2.775 km and then gradually
stretched to the top of the domain. The domain is sufficiently large to capture the effects of precipitation in the transition
from stratocumulus-to-cumulus clouds (Yamaguchi et al., 2017). We use the System for Atmospheric Model (SAM) version

6.10.10 to simulate the dynamics (Khairoutdinov and Randall, 2003) and the effects of radiation are computed using the Rapid
Radiative Transfer Model for Global Climate Models (RRTMG) using the two-stream approximation with extended vertical
profiles (Mlawer et al., 1997). In addition, we use a two-moment, bin-emulating bulk scheme to represent the microphysical
properties of the cloud system (Feingold et al., 1998).

The focus of this study is to assess the impact of aerosol perturbations on the SCT in the North-East Pacific region in a

warmer climate. We use the well-explored composite reference case from Sandu and Stevens (2011) to represent the present-
day (PD) scenario (JJA from 2005-2006). In this case, precipitation plays a prominent role in the onset of cumulus activity
and subsequent breakup (Yamaguchi et al., 2017; Prabhakaran et al., 2024). However, not all SCTs exhibit a key role for
precipitation (e.g., Bretherton et al. 2019). To suppress the precipitation-mediated transition we created an additional case with
reduced humidity in the free troposphere (FT). The humidity in the FT is lowered to 27% of the reference value and is referred

to as the dry FT case.

To represent the warmer climate at the end of the century (EoC), we consider two shared socioeconomic pathways (SSPs)
with forcings from the Representative Concentration Pathways (RCPs) from the recent coupled model intercomparison project
(CMIP6): (i) SSP1-2.6 represents sustainability and is one of the most optimistic warming scenarios, and (ii) SSP3-7.0 rep-
resents higher emissions associated with regional rivalry that maintains aerosol forcing (Riahi et al., 2017). The SSP1-2.6,

henceforth referred to as SSP1, represents a scenario with cleanup and warming (with respect to preindustrial) restricted to



| Case | [$CO_2$] | $\Delta$SST | $\omega$ | $N_a$ (mg$^{-1}$) |
|------|----------|-------------|----------|-------------------|
| PD | 1x | +0K | 1x (1.4x) | 150 |
| SSP3 | 2.3x | +3.1K | 0.90x (1.29x) | 150 |
| SSP1 | 1.2x | +0.9K | 0.96x (1.37x) | 120 |

**Table 1.** Large-scale conditions for all the simulations. The changes for each scenario are with respect to the values for the composite reference trajectory in Sandu and Stevens (2011). We report the increase in SST, and the factor by which $CO_2$ concentration ([$CO_2$]) and pressure velocity ($\omega$) changes. The values in the brackets for $\omega$ represent the conditions for the equivalent dry FT scenarios. $N_a$ shows the aerosol concentration in each scenario.

less than 2 °C in global mean temperature. The SSP3-7.0, henceforth referred to as SSP3, is considered the middle-of-the-road scenario. Note that the worst-case-scenario in CMIP6 is SSP5-8.5, however, this scenario is considered highly unlikely (Shiogama et al., 2023). Therefore, we use SSP1 and SSP3 as bounding scenarios to explore the efficacy of MCB in a warmer climate. We use the forcings from these scenarios to develop the equivalent EoC conditions for Sandu and Stevens (2011) using
the methodology described in Bretherton and Blossey (2014). A brief overview of the methodology is given below. We also included an additional 12 hour spin-up time for the cloud deck to adjust to its surroundings. During this time, the SST and the subsidence velocity remain constant. Aerosol perturbation (uniform surface injection) begins at the end of the spin-up period and is sustained for about 2.67 hours (approximate time for a ship to traverse the domain at a speed of 5 m s$^{-1}$).

To set up simulations under EoC conditions, the variables required from CMIP6 models are changes in SST ($\Delta$SST), $CO_2$
concentration, and changes in subsidence velocity. The SST and $CO_2$ concentrations are directly obtained from the global mean outputs from CMIP6. The key is to determine the sounding profiles for the scalars and the subsidence. The vertical temperature profile in the FT is close to that of a moist pseudoadiabat and is regulated by tropical-deep convective systems (Caldwell and Bretherton, 2009). Under warmer conditions, the temperature in the FT is obtained by following another moist pseudoadiabat with a temperature at sea level that is warmer than the present day value by $\Delta$SST. Similar to earlier studies, we
assume that RH in PD and EoC to be identical throughout the domain (Bretherton et al., 2013; Bretherton and Blossey, 2014). The vertical profile of the subsidence velocity (defined as the pressure velocity $\omega$) is determined by ensuring that the average FT temperature drift is minimal. To ensure this, subsidence-induced warming ($\omega \frac{\partial \theta}{\partial p}$) is balanced by clear-sky radiative cooling in the FT. For more details on determining subsidence profiles, see the appendix in Bretherton and Blossey (2014).

## 3 Results

### 3.1 Precipitation-mediated transition

Figure 1 shows the time evolution of the key cloud-topped boundary layer properties for the original reference case in Sandu and Stevens (2011) (red lines), henceforth referred to as the PD case, and its warmer-world counterpart at the EoC (black (SSP3) and green (SSP1) lines). We start by presenting the impact of warming on the SCT (solid lines) and then discuss the



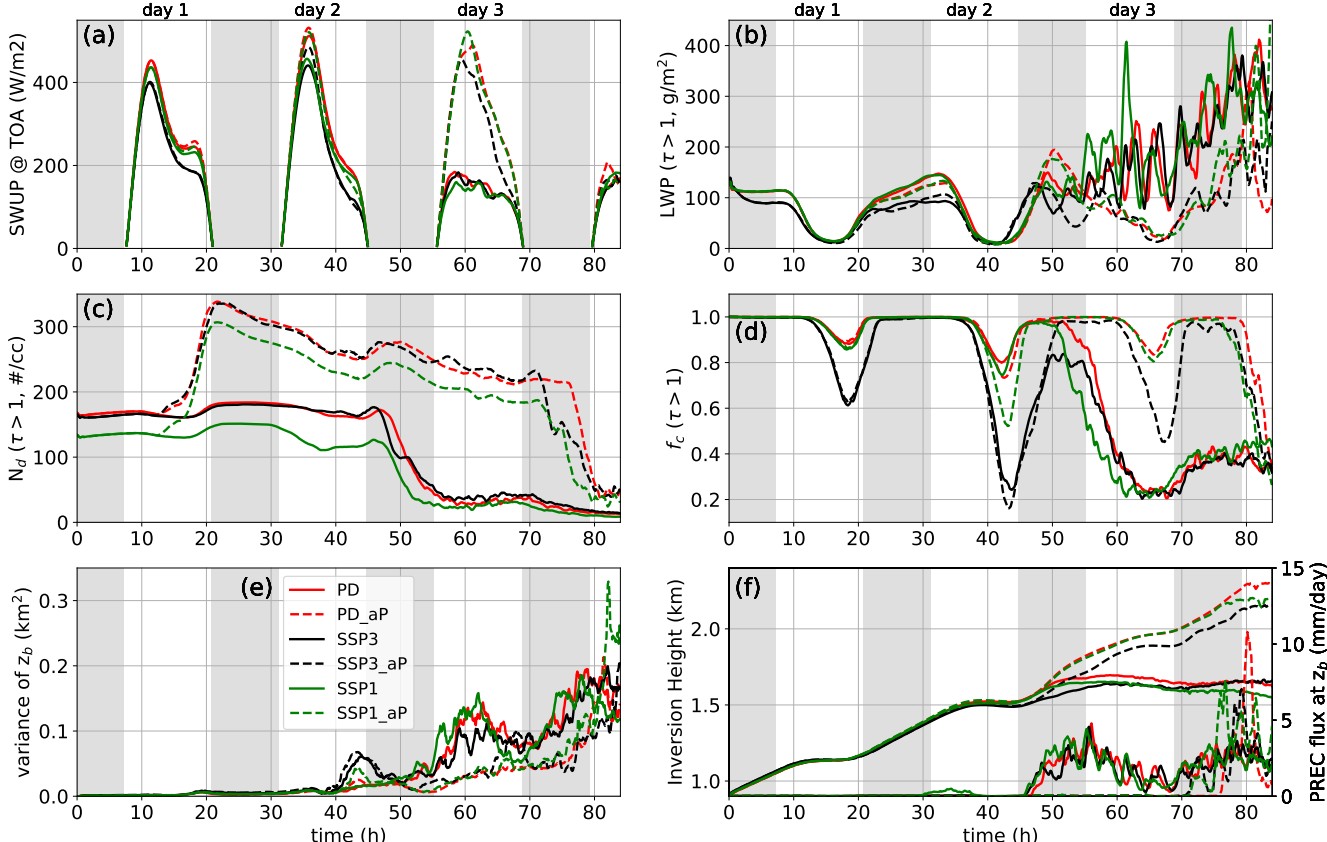

**Figure 1.** Time series of (a) short wave upward (SWUP) flux, (b) liquid water path (LWP), (c) cloud droplet concentration ($N_d$), (d) cloud fraction ($f_c$), (e) variance of cloud base ($z_b$), (f) domain-averaged precipitation flux (rain rate) at $z_b$ and domain-averaged height of the inversion layer ($z_i$) in the high humidity FT cases. $\tau$ is the cloud optical thickness used to identify the cloudy regions. The legend is shown in panel (e), and '_aP' in the legend identifies the cases with aerosol perturbation. The time period between sunset and sunrise is shaded in gray.

effect of aerosol perturbation (dashed lines). Note that SSP1 has fewer cloud droplets (120 mg$^{-1}$) compared to the other two
cases (150 mg$^{-1}$) and thus will be discussed separately. Additionally, we use a new metric for determining the onset of cumulus
activity. Earlier studies have used various metrics to define the magnitude of this transition. Sandu and Stevens (2011) defined
SCT based on reduction in scene albedo ($A$). In Prabhakaran et al. (2024) and Blossey et al. (2021), the transition is defined
on the basis of a sustained reduction (6 to 24 hours) in the cloud fraction $f_c$. The latter definition is applicable for the cases in
which precipitation plays a prominent role in the transition. Here, we use the spatial variance in the cloud base ($z_b$) as a metric
for the onset of cumulus clouds in the boundary layer. For nearly horizontally homogeneous stratocumulus clouds, the variance
in $z_b$ will be near zero. However, with the onset of cumulus activity, i.e., for cumulus-under-stratocumulus, the variance will
increase as the number of cumulus columns increases.





The evolution of PD (solid red line) is similar to the results in Prabhakaran et al. (2024). The diurnal cycle is clearly evident in the LWP and cloud fraction ($f_c$) time series (Fig. 1b, d). Note that cloud breakup in PD occurs earlier than in Prabhakaran et al. (2024). We attribute this to the additional 12-hour spin-up in the current simulations, which causes the boundary layer to deepen by an additional 200 m during this time period. The rapid reduction in $f_c$ (around 52 h in Fig. 1d) is triggered by the onset of strong precipitation at the cloud base (Fig. 1f), and is correlated with a rapid reduction in cloud droplet concentration $N_d$ (Fig. 1c). In the strongly warmer-world (SSP3), the LWP is substantially lower than its PD counterpart (Fig. 1b). Note that the entrainment velocity is also lower in SSP3, which is evident from the inversion layer height $z_i$ time series (Fig. 1f). This is due to weakening of the boundary layer turbulence due to climate change (Bretherton et al., 2013). $z_i$ in the warmer world is lower than in the PD scenario until the onset of precipitation below $z_b$ around 55 h. The reduction in LWP, despite the reduction in entrainment velocity, is an outcome of the entrainment-liquid flux (ELF) feedback associated with the increase in humidity jump across the inversion layer due to warming (Bretherton and Blossey, 2014). This increase in the humidity jump across the inversion layer enhances the evaporative cooling rate at the cloud-top, which translates to enhanced buoyancy production at cloud-top, thus increasing the entrainment flux despite a lower entrainment velocity (see Bretherton and Blossey (2014) for a detailed discussion of the ELF feedback mechanism). Additionally, $f_c$ undergoes a stronger diurnal cycle in a warmer climate, which is attributed to the strong reduction in cloud-top radiative cooling (due to the increase in $CO_2$ and water vapor concentration in the FT) as well as the ELF feedback (Bretherton and Blossey, 2014). Interestingly, the onset of cloud breakup in the warmer world (SSP3) around 50 h (Fig. 1d) is also driven by precipitation, and the timing of the onset of precipitation is a bit earlier than in PD. Post cloud breakup (after 64 h), the cloud properties and consequently the short wave upward (SWUP) flux for these two cases converge. The LWP and $f_c$ converge within a few hours ($\sim 5$ h), while the inversion heights take about 20 h to converge. Another variable of interest is the variance of the cloud base height $z_b$ (Fig. 1e). We see that cumulus activity is initiated earlier in SSP3 (enhanced value of the variance around 43 h), which is consistent with the earlier onset of precipitation. Note that the boundary layer coupling strengthens after sunset on day 2 and then weakens again after the onset of precipitation. Subsequently, in the remaining days, the magnitude of the $z_b$ variance continues to increase. Thus, apart from the onset of precipitation and the reduction in $f_c$, the variance in $z_b$ is also a good indicator of cumulus activity in this system.

In the muted warming scenario (SSP1), the effect of warming on cloud properties (solid green line in Fig. 1) is a lot weaker compared to SSP3. This is an obvious outcome of weaker forcing associated with climate change (see Table 1). However, there are additional contributions from the lower aerosol concentration. The LWP and $f_c$ are similar to the PD case on days 1 and 2. A possible explanation is that the effects of the ELF feedback associated with warming are offset by the lower entrainment rate associated with the lower aerosol concentration within the marine boundary layer, i.e., the reduction in LWP associated with the effects of climate change (warming and weakened radiative cooling) and the increase in LWP due to the reduction in $N_d$ (weakened sedimentation-entrainment and evaporative-entrainment feedbacks), offset each other. Here too, the lower aerosol concentration results in a slightly earlier onset of precipitation compared to PD. The evolution of cloud properties post-transition is similar to SSP3 but does not show any signs of convergence towards the PD values. $f_c$ and SWUP flux are consistently higher in SSP1 after the transition (after $\approx 65$ h).





Post aerosol injection ($\geq 12\,\mathrm{h}$) and prior to cloud breakup ($\lesssim 50\,\mathrm{h}$), the in-cloud ($\tau \geq 1$, where $\tau$ is cloud optical thickness) LWP is reduced relative to the unperturbed case (dashed lines in Fig. 1). The reduction in LWP is more substantial in PD and SSP1. In SSP3, a combination of weaker boundary layer turbulence (Bretherton et al., 2013) and lower LWP (Chen et al., 2024)
reduce the magnitude of the sedimentation-entrainment feedback. In all scenarios, $f_c$ is weakly affected by aerosol perturbation prior to cloud breakup ($\approx 50\,\mathrm{h}$) except in SSP1 around 42 h. The suppression of weak precipitation around sunrise on day 2 in SSP1 causes a reduction in $f_c$ between 40-45 h and causes cloud darkening around the same time, which is evident from the reduction in SWUP flux (green dashed line in Fig. 1a). A similar observation was reported in Prabhakaran et al. (2024). This precipitation-suppression induced darkening was attributed to the effects of enhanced SW absorption during the daytime.

The injection of aerosol into the boundary layer delays the onset of cloud breakup through precipitation suppression in PD and both warming scenarios. Despite the reduction in LWP, the suppression of precipitation maintains overcast conditions for an additional day in all scenarios, which leads to substantial cloud brightening on day 3. This is evident from the SWUP flux times series in Fig. 1a. The reduction of $f_c$ in the warmer world means that a reduced amount of cloud is available for brightening. This leads to a weaker SWUP flux enhancement in SSP3 on day 3 (Fig. 1a).

To better understand and quantify the impact of aerosol perturbation on cloud properties, we explore aerosol-induced changes to the SW cloud radiative effect (dCRE = CRE$_{\mathrm{unperturbed}}$ - CRE$_{\mathrm{aerosol\ perturbed}}$). A positive value of dCRE indicates cloud brightening. To decompose dCRE into contributions from $f_c$, N$_d$, and LWP we use the procedure laid out in Diamond et al. (2020) and Chun et al. (2023). The components of dCRE are written as

$$\mathrm{dCRE} = F_{in}\{\underbrace{f_c[(A_{c,\mathrm{N}_d,\mathrm{aP}} - A_c)]}_{\mathrm{dCRE-N}_d} + \underbrace{f_c[(A_{c,\mathrm{LWP,aP}} - A_c)]}_{\mathrm{dCRE-LWP}} + \underbrace{(f_{c,\mathrm{aP}} - f_c)(A_{c,\mathrm{aP}} - A_{clr})}_{\mathrm{dCRE}-f_c}\}, \tag{1}$$

where $F_{in}$ is the incoming solar radiation at top-of-the-atmosphere (TOA), $A_{clr}$ is the clear-sky albedo, $A_{c,\mathrm{N}_d/\mathrm{LWP}}$ is the cloud albedo contribution from changes in N$_d$ or LWP, $f_c$ is cloud fraction, and aP in the subscript indicates aerosol perturbation, respectively. dCRE-N$_d$ or dCRE-LWP represent the change in CRE while LWP or N$_d$ is unperturbed (with respect to aerosol), respectively. Both terms are directly proportional to the unperturbed $f_c$. dCRE-$f_c$ is a non-linear combination of cloud abedo changes and $f_c$ changes in response to aerosol perturbation. Figure 2 shows the dCRE and its decomposition to N$_d$, LWP, and
$f_c$ contributions for all three scenarios (PD, SSP1, and SSP3), and Table 2 shows the daytime average for the same quantities. Similar to Prabhakaran et al. (2024), we see that dCRE increases from day 1 to day 3 in all scenarios (Fig. 2a). On day 1, while the cloud is still adjusting to the aerosol perturbation, dCRE is highest for SSP1 and PD and lowest for SSP3 (Fig. 2a and Table 2). However, on day 2, the dCRE is highest for the SSP1 case and lowest for the PD case. On day 3, the highest dCRE is for SSP1 followed by PD and then SSP3 scenarios. These results suggest that the extent of warming and background aerosol
concentrations both play important roles in determining the aerosol-induced CRE changes. To gain a deeper understanding of these trends on each day, we examine the decomposition of dCRE (Fig. 2b,c, and d). On days 1 and 2, the SSP1 case has the highest dCRE-N$_d$. The albedo sensitivity scales as $A_c(1 - A_c)/(3\mathrm{N}_d)$ and thus saturates at higher N$_d$ (Platnick and Twomey, 1994). Consequently, the lower baseline N$_d$ in SSP1 results in a greater brightening potential. Additionally, on day 2 there is suppression of weak precipitation from the addition of aerosol in SSP1 (Fig. 1e), which further enhances the N$_d$
contribution due to a combination of Twomey and lifetime effects. The dCRE-N$_d$ contributions in the SSP3 and PD scenarios





**Figure 2.** Time series of changes in CRE (dCRE = $\text{CRE}_{\text{unperturbed}} - \text{CRE}_{\text{aerosol perturbed}}$) due to aerosol perturbation in the high FT humidity cases and its contributions from changes to $N_d$, LWP, and $f_c$. A positive value indicates cloud brightening. (a) dCRE, (b) $N_d$ contribution to dCRE, (c) LWP contribution to dCRE, (d) $f_c$ contribution to dCRE. $\tau$ is the cloud optical thickness. The legend is shown in panel (d). Note that y-axis range is different for each panel.

are similar, with a slightly higher value for PD (Table 2). On day 3, PD has the highest dCRE-$N_d$ contribution followed by SSP3 and SSP1. Note that on day 3, precipitation at $z_b$ is suppressed due to aerosol perturbation in all scenarios and leads to substantial brightening (Fig. 2a). In dCRE-LWP, we see that the net contribution is negative for all scenarios on days 1-3 due to the reduction in LWP from enhanced entrainment rate (Fig. 2c). On days 2 and 3, the highest magnitude is for PD followed by SSP1 and SSP3. On day 3, dCRE-$N_d$ and dCRE-LWP contributions are comparable and thus cancel each other in all three scenarios, and the dominant contribution is from dCRE-$f_c$ due to precipitation suppression (Fig. 2d). After precipitation suppression (day 3), the increase in $f_c$ is directly proportional to the enhancement in dCRE. During this time, the net increase in dCRE is the lowest for SSP3 and the highest for both SSP1 and PD (Table 2). Note that it is tempting to interpret these trends



as an outcome of changes in ACI due to warming. Instead, these trends are a combination of the warming-induced changes in
LWP and $f_c$, and aerosol-induced changes. A more detailed discussion is presented in Section 4.2.

## 3.2 Entrainment-driven transition

Figure 3, similar to Fig. 1, shows the time series of key boundary layer properties in all scenarios with dry FT humidity
(entrainment-driven transition). In this case, precipitation does not play a dominant role in the transition to a cumulus-topped
boundary layer. This is evident from Fig. 3f, which shows that the precipitation flux at $z_b$ is negligible prior to the onset of
cumulus activity (around 66 h) in all unperturbed scenarios. As noted previously, we use the variance in $z_b$ as a metric to assess
the onset of cumulus activity.

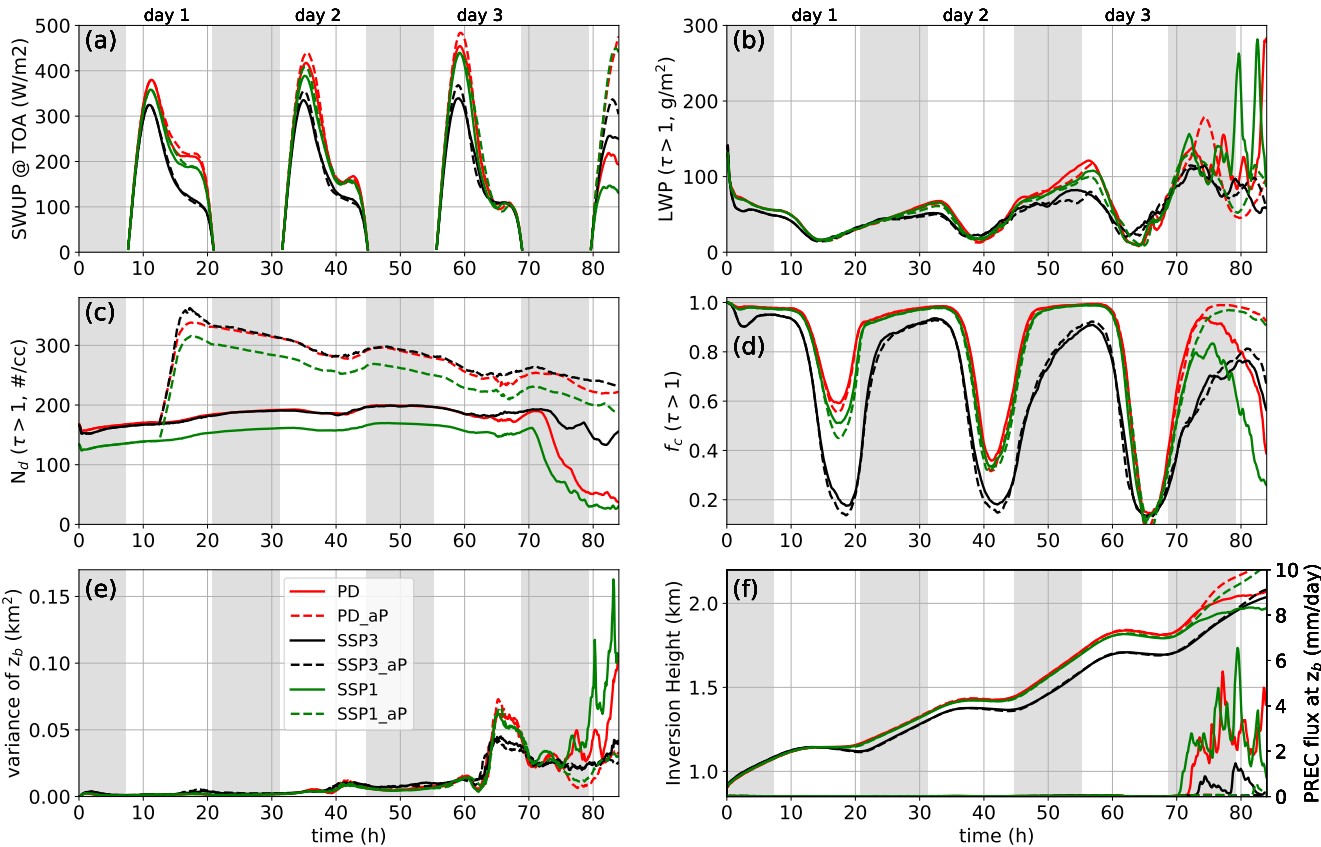

**Figure 3.** Same as Fig. 1, but for the dry FT cases.

The reduction in FT humidity substantially lowers the cloud LWP (Fig. 3b) compared to the corresponding cases in Fig. 1.
Over time, the increase in SST enhances the LWP in the cloud layer in all the unperturbed cases. Similar to the cases in the
precipitation-mediated scenario, the LWP is lowest in the SSP3 case until the onset of precipitation, and similar in the PD and
SSP1 cases on day 1 and most of day 2. From the end of day 2, the LWP in PD is greater than in SSP1. On day 3, the presence





of precipitation at $z_b$ in the PD and SSP1 scenarios results in fluctuations in the LWP time series. The relatively low LWP in SSP3 results in weak precipitation below the cloud base until the end of the simulation. The formation of precipitation and its intensity on the final day is also evident from the rate of decrease in $N_d$ in Fig. 3c. The $f_c$ time series is substantially altered by the degree of climate change (Fig. 3d). The increase in SST due to climate change and the weakening of radiative cooling at

cloud-top, enhances the magnitude of changes in $f_c$ over the course of a diurnal cycle. Consequently, SSP3 has the lowest $f_c$, followed by SSP1 and the reference case. It is interesting to note that the minimum value in $f_c$ is similar on all three days in SSP3, while, in the SSP1 and PD scenarios, the minima in $f_c$ decrease from day 1 to day 3. On day 3, the value of the minimum in $f_c$ is similar in all three cases around 66 h, marking the onset of cumulus clouds. This is evident from the sharp increase in the cloud base variance time series ($\approx 66$ h in Fig. 3e). The onset of precipitation in the final 12 hours of the simulation results

in cloud breakup, lowers $f_c$ and $z_b$, and further increases the variance in $z_b$ (Figs. 3 d, e, and f). Note that despite the similarity in the onset times of cumulus clouds across scenarios, the cloud breakup timings are quite different. Substantial precipitation occurs first in SSP1 due to a lower $N_d$, followed by PD, and then SSP3 (Fig. 3f). Substantially lower LWP in SSP3 results in a delayed and weaker precipitation flux (20-50% lower compared to PD). Consequently, cloud breakup is slower in SSP1 (Fig. 3 d) and the SWUP flux is highest for SSP1 on day 4 (Fig. 3 a).

The injection of aerosol lowers the LWP in all three scenarios until the onset of precipitation (dashed lines in Fig. 3b). The magnitude of the change in LWP increases with time until the onset of precipitation on the final day (Fig. 3b). Additionally, during the daytime, $f_c$ also decreases weakly due to the increase in aerosol concentration prior to the onset of precipitation (Fig. 3d). Notable effects related to aerosol perturbation are evident only post precipitation suppression. The suppression of precipitation on the final day enhances $f_c$ in PD and SSP1 scenarios. Note that in the SSP3 scenario, the increase in $f_c$ post

aerosol injection on the final day is lower compared to SSP1 and PD due to the weaker precipitation flux at $z_b$. Furthermore, the addition of aerosol does not influence the onset of cumulus clouds (see the variance of $z_b$ time series in Fig. 3e).

Figure 4 shows the changes in SW CRE due to aerosol perturbation in the entrainment-driven transition cases. Until the onset of precipitation (day 3/day 4), the enhancement in CRE is weaker compared to the high-humidity cases (Fig. 4 a). This is an outcome of lower LWP and $f_c$. However, post precipitation onset on the final day, the enhancement in CRE is comparable to the moist FT case. The dCRE is comparable in PD and SSP1 on days 1 and 2. On day 3, the dCRE in SSP1 is substantially

higher than in PD. For SSP3, the dCRE values are substantially lower than the other two cases on all days. The decomposition of dCRE offers further insights into the evolution of dCRE across scenarios. The $N_d$ contribution in SSP3 is the lowest on all three days, and the values are similar in SSP1 and PD on each day with slightly higher values for the SSP1 case due to its lower baseline $N_d$ (Fig. 4 b). The dCRE-LWP and dCRE-$f_c$ contributions are largely negative with similar magnitude in both SSP1

and PD in the first two days (Fig. 4 c and d). Additionally, the lowest magnitude for the dCRE-LWP on each day is for SSP3. On day 3, there are strong fluctuations in dCRE-LWP and dCRE-$f_c$ due to strong cumulus activity (Fig. 4 c, d). The substantially higher dCRE on day 3 for SSP1 is the outcome of strong positive LWP and $f_c$ contribution due to cumulus activity which is stochastic in nature and thus cannot be directly attributed as a response to aerosol perturbation. A more detailed discussion on ACI after separating the effects of climate change will be presented in Sec. 4.2.



**Figure 4.** Time series of changes in CRE due to aerosol perturbation in the dry FT cases and its contributions from changes to $N_d$, LWP, and $f_c$. (a) dCRE, (b) $N_d$ contribution to dCRE, (c) LWP contribution to dCRE, (d) $f_c$ contribution to dCRE. $\tau$ is the cloud optical thickness. The legend is shown in panel (d).

## 4 Discussion

In the previous section, we explored the effects of climate change on SCT and ACI in the North-East Pacific region. We analyzed two cases of SCT (i) a precipitation-mediated transition, and (ii) an entrainment-driven transition. These cases were created by altering the humidity in the FT, with the high humidity case [the reference case in Sandu and Stevens (2011)] resulting in strong precipitation and the low humidity case resulting in low or negligible precipitation. These cases were subjected to different climate and aerosol perturbations. The results show that until the onset of precipitation the effect of aerosol perturbation is much weaker than the effects of climate change. However, in precipitating cloud systems, the effects of aerosol-induced precipitation suppression dominate over the effects of climate change. In addition, we explore the impact of climate change on the onset of cumulus activity and cloud breakup. The results show that the onset of cumulus activity is





| case | ID | day 1 (W m$^{-2}$) | | | | | day 2 (W m$^{-2}$) | | | | | day 3 (W m$^{-2}$) | | | | |
|---|---|---|---|---|---|---|---|---|---|---|---|---|---|---|---|---|
| | | dCRE | $N_d$ | LWP | $f_c$ | RES | dCRE | $N_d$ | LWP | $f_c$ | RES | dCRE | $N_d$ | LWP | $f_c$ | RES |
| Moist | PD | 3.7 | 4.9 | -2.6 | 0.2 | 1.2 | 3.8 | 22.4 | -24.9 | -0.1 | 6.4 | 177.0 | 39.0 | -36.6 | 148.4 | 26.1 |
| | SSP1 | 4.3 | 6.0 | -2.8 | 0.2 | 1.0 | 16.9 | 32.6 | -14.6 | -1.6 | 0.6 | 188.3 | 31.8 | -30.6 | 173.5 | 13.6 |
| | SSP3 | 2.6 | 3.9 | -3.4 | 0.6 | 1.5 | 9.0 | 18.3 | -5.2 | -2.0 | -2.1 | 137.2 | 33.6 | -32.0 | 120.1 | 15.5 |
| Dry | PD | 5.7 | 10.4 | -3.9 | -1.5 | 0.7 | 6.8 | 15.7 | -6.3 | -2.9 | 0.3 | 5.6 | 10.3 | -4.3 | -1.5 | 1.1 |
| | SSP1 | 4.1 | 10.9 | -3.8 | -2.7 | -0.2 | 7.2 | 17.0 | -6.3 | -3.5 | -0.1 | 13.3 | 11.1 | 5.0 | 3.2 | -6.0 |
| | SSP3 | 0.3 | 5.1 | -1.9 | -2.7 | -0.1 | 1.9 | 11.5 | -5.1 | -4.9 | -0.3 | 2.0 | 8.1 | -5.9 | -1.5 | 1.2 |

**Table 2.** Cloud radiative effect enhancement (dCRE = CRE$_{\text{baseline}}$ - CRE$_{\text{aerosol perturbation}}$) and its decomposition at the TOA for all cases. The budgeting is done using cloud properties for a threshold of $\tau > 1$. The quantity for each day is averaged between sunrise and sunset. Column RES is the residual of the dCRE budget. RES = dCRE - sum of dCRE components. A smaller value for RES suggests a uniform cloud field. Note that a positive value indicates cloud brightening.

not affected by climate change under dry FT conditions and occurs around noon on day 3 (around 66 h). However, the cloud
breakup times are different, with the earliest in SSP1, followed by PD and then SSP3. In the moist FT case, SSP3 results in an early onset of cumulus activity (around 42 h) and cloud breakup (around 45h).

Although this paper focuses on understanding the impact of climate change on ACI in marine-low clouds, it is also important to quantify the effects of climate change on the unperturbed (with respect to aerosol) state of the cloud. This will aid us in separating the effect of aerosol perturbation from that of climate change. In the next subsections, we discuss the impact of
climate change on the unpertubed (aerosol) state of the cloud, followed by ACI in a warmer climate, and finally the impact of climate change and aerosol perturbation on SCT and MCB.

### 4.1 CRE: Role of climate change

The impact of climate change on CRE in cases with moist and dry humidities is shown in Fig. 5, which depicts the difference in CRE between PD conditions and EoC conditions (dCRE$_{\text{clm}}$). A positive value indicates cloud brightening. Note that dCRE$_{\text{clm}}$
is different from dCRE in Eq. 1, which represents the change in CRE due to aerosol perturbation. dCRE$_{\text{clm}}$ only considers the cases without aerosol perturbation. In the moist FT case, the magnitude of the CRE decreases (cloud darkening) relative to its PD values before the onset of strong precipitation in both scenarios. This is associated with the reduction in LWP and $f_c$ through the effects of warming SST (due to climate change) and the weakening of radiative cooling at cloud-top (Bretherton and Blossey, 2014). This is consistent with the positive low-cloud feedback obtained from other LES studies (Bretherton et al.,
2013; Bretherton and Blossey, 2014) and observational studies (Myers et al., 2021; Ceppi et al., 2024; Cesana and Del Genio, 2021; Klein et al., 2017). As expected, to leading order, the reduction in CRE is stronger for the SSP3 case. Note that dCRE$_{\text{clm}}$ in SSP1 is comparable to SSP3 on day 2. This is due to weak precipitation in SSP1 (Fig. 1f) resulting in a reduction in LWP and




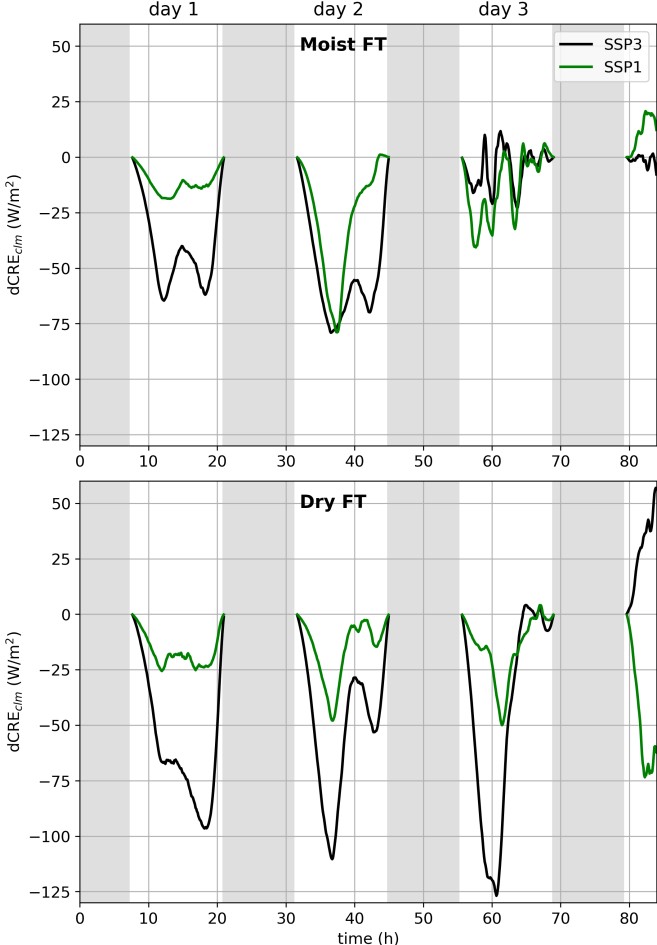

**Figure 5.** Changes in CRE due to climate change. The color code is same as in Figs. 1 and 3. The panel at the top (bottom) represents the moist (dry) FT case. Note that the changes are relative to the PD scenario without aerosol perturbation. A negative value indicates cloud darkening due to climate change.

$N_d$ between 30 and 40h in Fig. 1 b and c. On day 3, after the cloud breakup is initiated, the magnitude of $dCRE_{clm}$ decreases substantially in both scenarios, so that by the end of day 3 and day 4 $dCRE_{clm}$ is very low. Note that SSP1 has a slightly
higher magnitude on day 3 as precipitation starts earlier owing to a lower aerosol concentration. Furthermore, we also see that $dCRE_{clm}$ is positive on day 4 in SSP1 owing to higher $f_c$.

In the dry FT case, precipitation does not play a prominent role until after sunset on day 3. Until then, the magnitude of $dCRE_{clm}$ is highest for SSP3. After the onset of cumulus activity ($\geq$ 65 h on day 3), the magnitude of $dCRE_{clm}$ in both scenarios reduces to near zero. However, the slow cloud breakup in SSP3 due to low LWP results in a positive value for
$dCRE_{clm}$. Interestingly, the dry FT case exhibits larger (negative) $dCRE_{clm}$ compared to the moist FT case and therefore




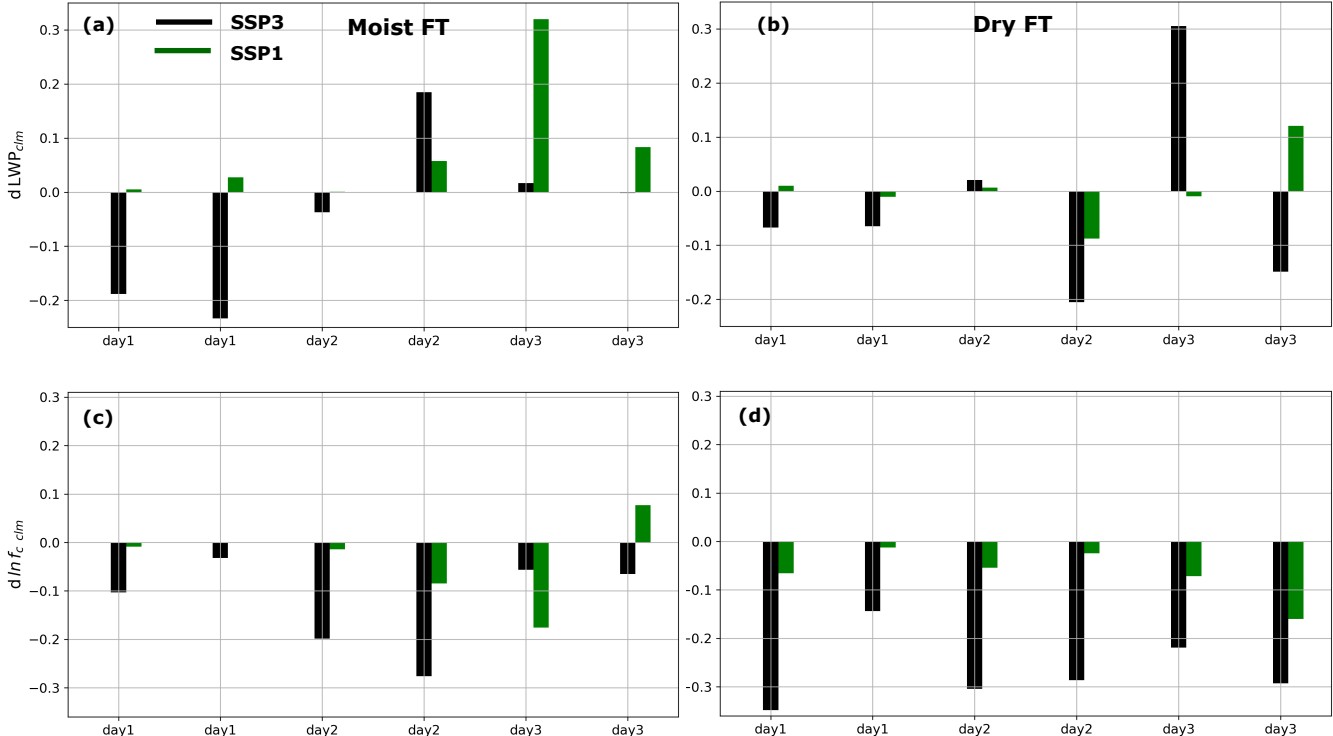

**Figure 6.** Fractional changes in LWP (top) and $f_c$ (bottom) due to climate change. The color code is shown in panel (a) and is same as in Figs. 1 and 3. The panels to the left represent the moist FT case and the panels to the right represent the dry FT case. Note that the changes are relative to the equivalent present day scenario. These are average values for each day. Two values per day represent the two averaging periods, between sunrise and sunset (first data point), and sunset to sunrise (second data point).

warrants further scrutiny. In particular, we need to understand the impact of climate change on $f_c$ and LWP, which is provided below.

Figure 6 depicts the fractional changes in LWP and $f_c$ due to climate change. While we show the results for both SSP1 and SSP3, for ease of discussion we focus on SSP3. In the scenarios with moist FT, the fractional reduction in $f_c$ and LWP is

substantial. On day 1, the reduction in $f_c$ ranges between 5 and 10% and increases to 20-25% on day 2. The reduction in LWP is around 20% on day 1 and less than 5% on day 2. (Note that we are only considering the time period before the onset of precipitation, i.e., before sunset on day 2.) In the dry FT case, the reduction in $f_c$ is higher than its moist counterpart (around 30 to 35% on days 1 and 2). However, the reduction in LWP is lower compared to the moist case (less than 10% on day 1). These results indicate that the extent of reduction in $f_c$ due to climate change is a function of LWP. These trends in $f_c$ and LWP

show the stronger effect of climate change on CRE in the dry FT case. We attribute this to the shielding effect of higher LWP in the moist FT case. In all scenarios in Fig. 1 and Fig. 3, a substantial reduction in $f_c$ does not start until the LWP reaches a sufficiently low value (Prabhakaran et al., 2023). In simpler terms, cloud thinning precedes reduction in $f_c$. Consequently,





the higher LWP in the moist FT case shields $f_c$ from the effects of weakened radiative cooling and climate-change induced warming. The lower LWP in the dry FT case results in stronger reduction in $f_c$. Since CRE, to leading order, is controlled by

$f_c$, until the onset of precipitation, dCRE$_{\text{clm}}$ is higher in the dry FT cases.

In the moist FT case, after cloud breakup on day 3, the fractional changes in $f_c$ (around 5%) and LWP (around 1-2%) are low, resulting in low dCRE$_{\text{clm}}$ (Fig. 5). After the precipitation-induced breakup, the cloud layer transitions into an open-cellular structure and has the characteristics of a surface driven cloud system (e.g., cumulus clouds) (Wang and Feingold, 2009). Thus, the low value for dCRE$_{\text{clm}}$ post cloud breakup is consistent with the locally weak low-cloud feedback seen in marine cumulus

clouds (Bretherton et al., 2013; Vogel et al., 2016; Dagan et al., 2018; Myers et al., 2021).

### 4.2 CRE: Aerosol-cloud interaction

Changes in SW CRE due to aerosol injection (dCRE) are shown in Figs 2 and 4, and the daytime average values are shown in Table 2. The decomposition of dCRE into individual contributions from N$_d$, LWP, and $f_c$ is given by Eq. 1. The Twomey contribution dCRE-N$_d$ and the LWP contribution dCRE-LWP to dCRE are both proportional to unpertured (with respect to

aerosol) $f_c$. Furthermore, these components of dCRE are proportional to changes in the cloud albedo due to N$_d$ (or LWP) while LWP (or N$_d$) is held constant (unperturbed state). See Wood (2021) for additional details. The $f_c$ contribution, dCRE-$f_c$, is a more complex component of dCRE. It is a non-linear function of changes in $f_c$ and total cloud albedo post aerosol perturbation. Under moist FT conditions, the magnitude of dCRE-$f_c$ is negligible until the onset of precipitation (see first two days in Fig. 2). Then, to leading order, in the absence of substantial $f_c$ changes, dCRE can be expressed as the sum of dCRE-N$_d$

and dCRE-LWP

$$\text{dCRE} = F_{in}\{\underbrace{f_c[(A_{c,\text{N}_d,\text{aP}} - A_c)]}_{\text{dCRE}_{Nd}} + \underbrace{f_c[(A_{c,\text{LWP,aP}} - A_c)]}_{\text{dCRE}_{\text{LWP}}}\}. \tag{2}$$

If we factor out $F_{in}$ and $f_c$ from the above expression, we obtain

$$\frac{\text{dCRE}}{F_{in}f_c} = [(A_{c,\text{N}_d,\text{aP}} - A_c)] + [(A_{c,\text{LWP,aP}} - A_c)]. \tag{3}$$

The first term on the right side of Eq. 3 is the change in cloud albedo due to the changes in N$_d$ while LWP is kept unchanged

and the second term is the change in cloud albedo due to the changes in LWP while N$_d$ is kept unchanged.

The components of albedo changes based on Eq. 3 are shown in Fig. 7 for moist and dry FT cases. There is a remarkable collapse for the N$_d$ contributions across different scenarios in both cases (Fig. 7 a and b). We see minor differences between SSP1 and the other two scenarios, which we attribute to the lower aerosol concentration in SSP1 and associated entrainment feedbacks. In the moist FT case, we see a strong deviation in the SSP1 scenario on day 2 due to suppression of weak precipita-

tion. Apart from these minor differences, the collapse in the N$_d$ albedo component extends to clouds with strong precipitation as well (days 3 and 4). In the dry FT case, we see a substantial spread in the N$_d$ albedo component between the scenarios only on day 4. This is related to differences in the timing of the onset and intensity of precipitation across scenarios (Fig. 3f).

The LWP albedo contribution time series is quite noisy and the collapse across different scenarios is less remarkable compared to the N$_d$ component (Fig. 7 c and d). Note that unlike N$_d$, the LWP adjustment to aerosol perturbation is not instantaneous





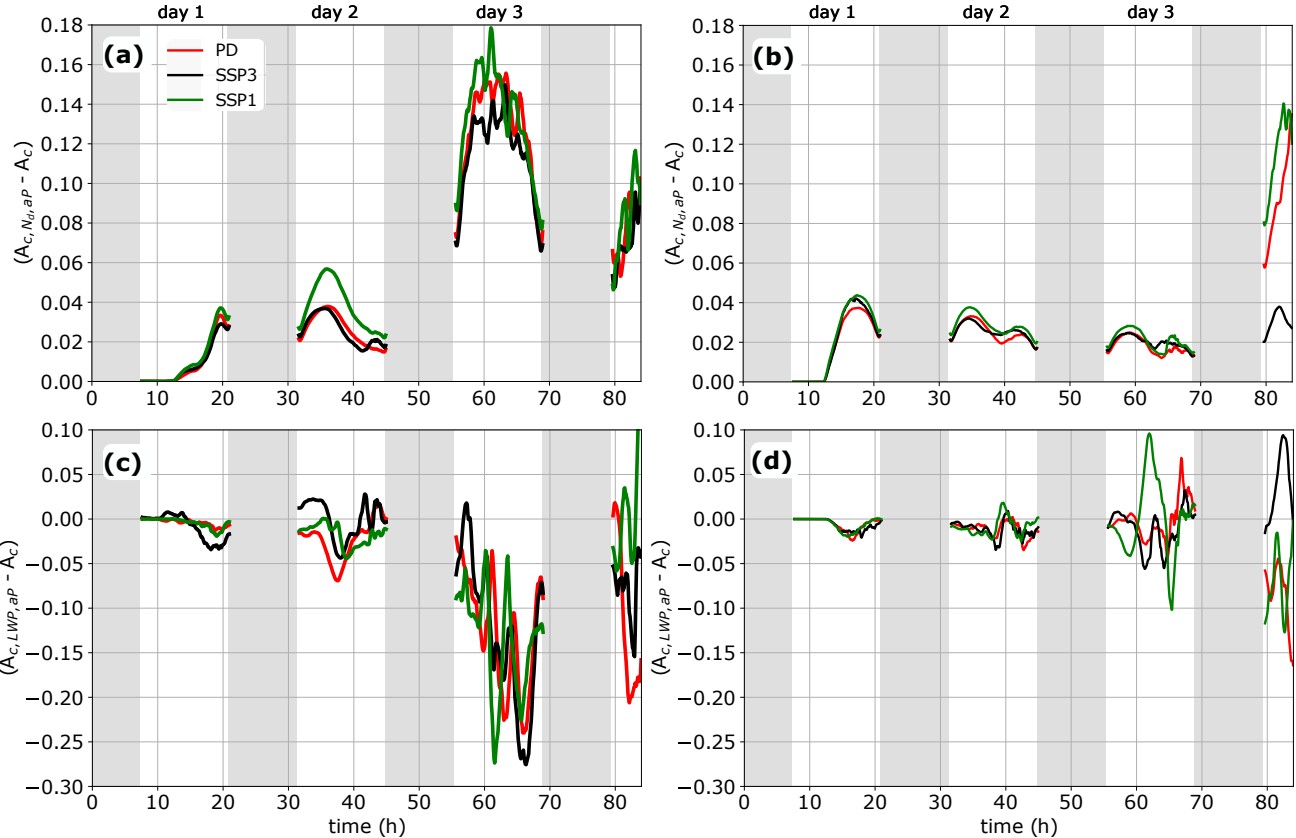

**Figure 7.** Components of cloud albedo changes due to aerosol perturbation in the moist (left column) and dry (right column) cases. The panels at the top (bottom) show the $N_d$ (LWP) component in Eq. 3. The color code is shown in panel (a).

and could range from a few to several hours (Chen et al., 2024). The critical role of meteorology in determining these time scales is poorly understood. These differences in the LWP adjustment timescale could explain the better collapse in the LWP component of the albedo changes in the scenarios with dry FT compared to the equivalent moist FT scenarios in the first two days. Nevertheless, to leading order, there is reasonable collapse across scenarios in both cases on the first two days and the collapse extends to the third and fourth day in the moist FT scenarios. In the dry FT case, we see a substantial spread across

scenarios on days 3 and 4 due to the onset of cumulus activity and precipitation, respectively (Fig. 7d).

For each humidity case, the similarity in the values for $N_d$ and LWP components of albedo change across climate scenarios is, at first sight, puzzling as the impact of climate change on LWP and, by extension, the cloud albedo is clearly evident from the time series plots in Figs. 1b and 3b. It is important to note that the bulk of the reduction in cloud LWP due to climate change is visible post-sunset and before sunrise. Interestingly, after sunrise, the impact of climate change on cloud LWP decreases as

clouds with higher LWP are more susceptible to SW absorption (Petters et al., 2012). During this time, the dominant effect of





climate change on CRE is evident from the changes in $f_c$ (Sec. 4.1). Thus, by factoring out $f_c$ in Eq. 2 we are eliminating (to leading order) the effects of climate change on dCRE in non-precipitating marine low-clouds.

The similarity we see here appears to be case-specific as there is no universality among these albedo change time series between the moist and dry FT cases. This local similarity suggests that in stratocumulus clouds, changes in albedo contributions from $N_d$ and LWP are immune to climate change, i.e., the magnitude of Twomey effect and LWP adjustments do not change within the cloudy region. The scenario-specific differences we see in the dCRE components in Table 2 and Figs. 2 and 4 are an outcome of the warming-induced changes to the unperturbed state (mainly $f_c$) of the cloud system and not an outcome of the impact of climate change on ACI.

We have established that the $N_d$ and LWP components of $\frac{\mathrm{dCRE}}{f_c}$ are, to leading order, invariant under climate change. If we know the magnitude of the albedo adjustment in the non-precipitating stratocumulus clouds in the present day, then we can estimate the magnitude of CRE adjustment in a warmer climate provided we know $f_c$ under warmer conditions. Bretherton et al. (2013) have established that changes in the CRE of stratocumulus clouds due to planetary warming can be approximated as a linear combination of changes in meteorological variables or cloud-controlling factors. This linear approximation has also been used in several recent observational studies (Cesana and Del Genio, 2021; Myers et al., 2021; Ceppi et al., 2024; Klein et al., 2017). We extend this to account for cloud droplet concentration and express changes in CRE in non-precipitating clouds under climate change as

$$dCRE = \sum_{i=1}^{n} \frac{\partial \mathrm{CRE}}{\partial M_i} dM_i + \frac{\partial \mathrm{CRE}}{\partial N_d} dN_d, \tag{4}$$

where $M_i$ represents $i^{\mathrm{th}}$ cloud-controlling meteorological factor and $\frac{\partial \mathrm{CRE}}{\partial N_d} = F_{in} f_c \frac{\partial A_c}{\partial N_d}$. As was discussed earlier, $\frac{\partial A_c}{\partial N_d}$ is invariant under climate change. In precipitating systems, the addition of aerosol invokes a non-linear response to CRE through adjustments in $f_c$ and $A_c$. To represent precipitation-related changes to dCRE, additional higher-order terms are required in the Taylor series expansion of dCRE in Eq. 4. A more detailed study is required to address this issue.

### 4.3 Stratocumulus-to-cumulus transition

It is important to note that the low-cloud transitions investigated here are (i) precipitation-mediated (moist FT): from a closed marine startocumulus cloud deck to an open cellular low-cloud layer, and (ii) entrainment-driven (dry FT): the transition from a marine stratocumulus cloud deck to a cumulus-under-stratocumulus cloud deck. Technically, in both cases, the transition to a cumulus-topped boundary layer is not yet complete.

In the precipitation-mediated transition, climate change affects the onset of the transition. The transition is initiated earlier in SSP3 than in the PD case. Note that both cases have similar aerosol number concentration at the start, and so microphysics is unlikely to explain the difference in initiation time. The boundary layer decoupling is stronger in the SSP3 scenario compared to the PD scenario due to weakened radiative cooling at cloud-top and increased surface latent heat flux (Bretherton and Wyant, 1997). This is also supported by the buoyancy integral ratio (BIR) time series in Fig. 8a. BIR is a ratio between the magnitudes of the negative buoyancy flux in the sub-cloud layer and the positive buoyancy flux in the boundary layer (Turton and Nicholls,



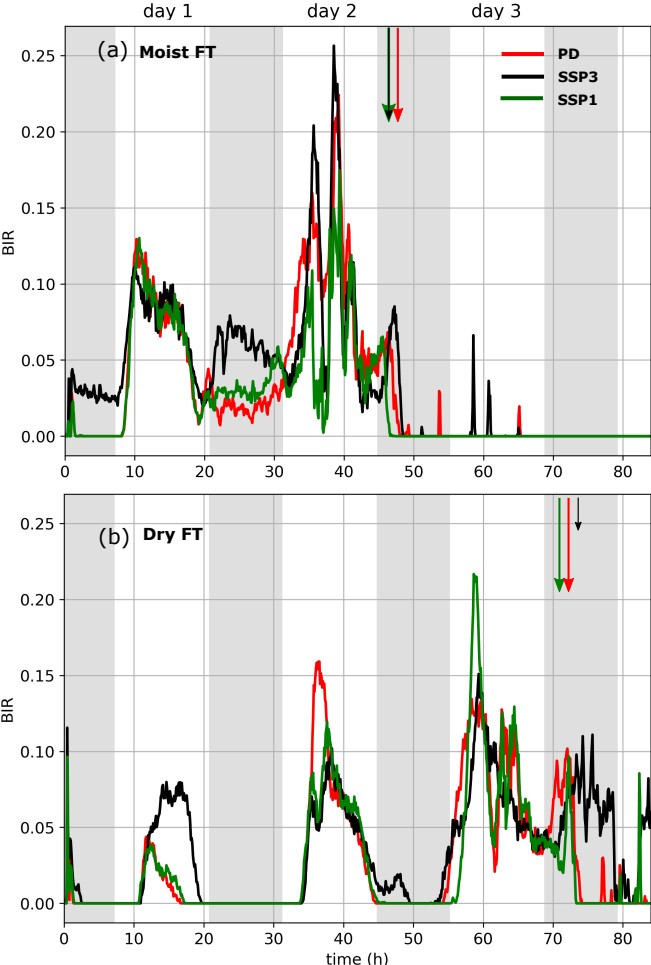

**Figure 8.** Buoyancy integral ratio (BIR) for (a) moist FT scenarios, and (b) dry FT scenarios. The color code is shown in panel (a). The vertical arrows at the top of each panel represents the cloud base precipitation onset time. The length of these arrows is a relative measure of the precipitation intensity.

1987; Bretherton and Wyant, 1997). A higher BIR value indicates a stronger decoupling. Note that the level of decoupling is higher in SSP3 especially during the night on day 1. The magnitude of decoupling increases in strength on day 2 which
eventually leads to the earlier onset of cumulus activity on day 2. The stronger decoupling in SSP3 is also evident from the jump in the magnitude of the cloud base variance around 43h (Fig. 1e). The earlier onset of cumulus activity eventually leads to the early onset of precipitation-induced breakup. An earlier transition also occurs in the SSP1 case, but this is driven by both the lower aerosol concentration and the effects of climate change.

    In the entrainment-driven case, the effects of climate change do not influence the onset of the cumulus clouds. However, the
onset of breakup, which is influenced by precipitation, is affected by the effects of warming through the changes in LWP. The





substantial reduction in LWP in the SSP3 scenario delays the onset and weakens the intensity of precipitation. This results in brighter clouds in the SSP3 case post-transition. This is in contrast to the behavior prior to the transition. It is puzzling that unlike in the scenarios with moist FT, climate change does not affect the onset of cumulus activity in the dry scenarios despite changes in the cloud properties. A possible explanation for this could be that the lower humidity in the FT ensures that radiative

cooling at the cloud top is large enough to preserve the structure of the marine boundary layer (MBL) despite the effects of climate change. The BIR time series in Fig. 8b for the dry FT case shows that there is a certain degree of decoupling during the daytime on the first two days and the extent of decoupling increases from day 1 to day 2. However, after sunset, the boundary layer is recoupled in all scenarios during this time period. On day 3, the decoupling gets a lot stronger in all scenarios, which eventually results in cumulus activity. Note that in the scenarios with moist FT a certain level of decoupling is maintained in

the boundary layer even after sunset on all days until the onset of precipitation. Comparison of the BIR time series in both cases supports the hypothesis that substantially lower radiative cooling of the boundary layer in the moist FT case (83 W m$^{-2}$ in moist FT and 113 W m$^{-2}$ in dry FT) makes the onset of cumulus activity more susceptible to climate change. However, more case studies are required to assess the robustness of this conclusion.

The injection of aerosol delays the breakup of the stratocumulus deck in both moist and dry FT cases by delaying the onset

of precipitation in cumulus-cloumns. This leads to substantial brightening of the cloud deck in all climate scenarios. However, the onset of cumulus activity in both cases is largely unaffected by aerosol perturbation (all scenarios in dry FT and SSP3 in moist FT) unless influenced by precipitation formation (PD and SSP1 in moist FT).

## 4.4 Marine cloud brightening

Figure 9 shows time-averaged SWUP flux (panels to the left) and the long-wave upward flux (LWUP) (panels to the right) at

the TOA for all cases simulated in this study. There are two data points for each day, each representing the average values between sunrise and sunset and between sunset and sunrise. The top panels represent the moist FT case and the bottom panels represent the dry FT case. As was concluded from the discussion in the previous subsection, the effect of climate change tends to dominate over the effects of aerosol perturbation in the absence of precipitation (solid bars in Fig. 9). In the SSP1 and SSP3 scenarios, the enhancement in SWUP flux due to aerosol perturbation (difference between hatched bars and solid bars)

is smaller than the reduction associated with climate change. The effects of positive cloud feedback reduce the brightening prospects of marine low-clouds in the absence of precipitation (see Sec. 4.1 for a detailed discussion).

Suppression of precipitation leads to substantial enhancement in SWUP flux. Some of this enhancement is offset by the reduction in the outgoing LW radiation due to a rise in cloud top height (see the panels to the right in Fig. 9, and Fig. 1f). Despite this, in all climate scenarios, suppression of precipitation leads to an enhancement in SWUP flux that is substantially

greater than the unperturbed PD scenario, as the effects of cloud feedback are weaker after the cloud breaks up to open-cellular clouds. However, the net enhancement in SWUP flux due to precipitation suppression is not immune to climate change. For instance, in the moist FT case, the highest enhancement in $\overline{SWUP}$ is for the PD and SSP1 scenarios due to the greater $f_c$ during daytime on day 3 (Fig. 9 a).





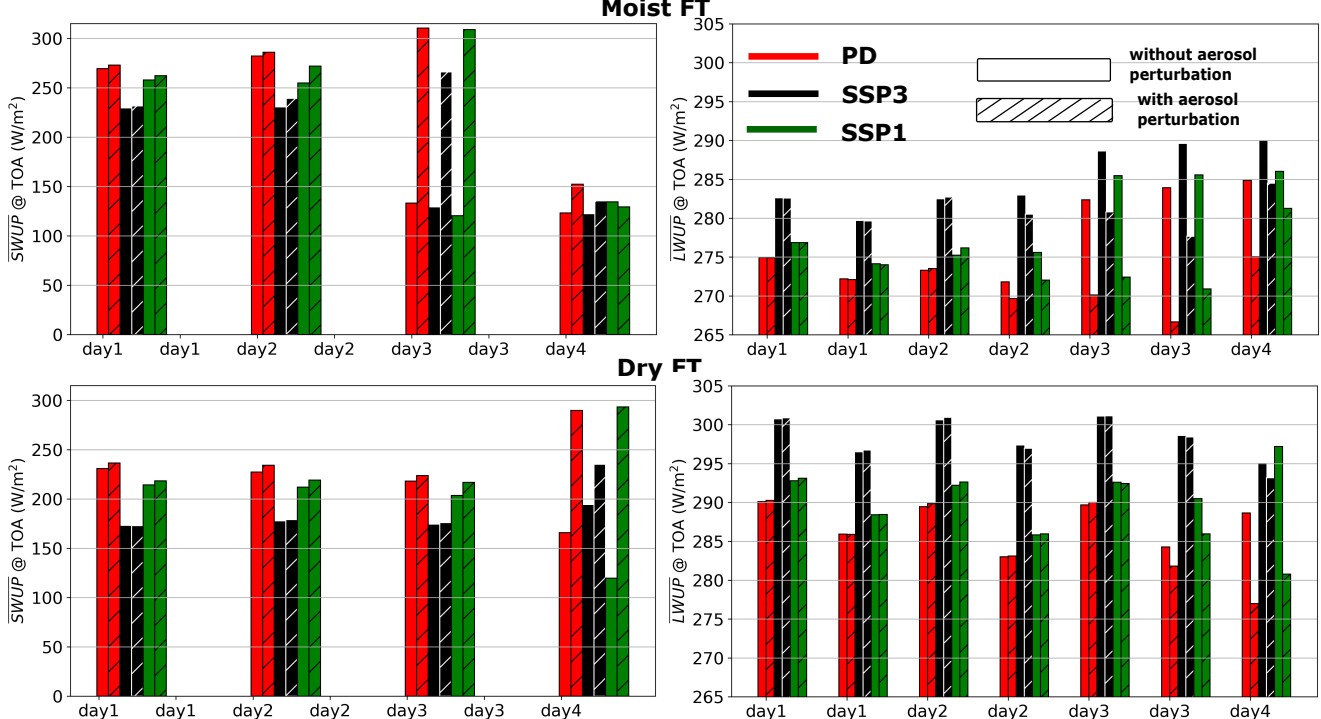

**Figure 9.** Bar plots representing average SWUP (panels to the left) and LWUP (panels to the right) flux in the TOA for each scenario in the moist (top panels) and dry (bottom panels) FT cases. For each day, two average values are reported per scenario. The averaging period for each data point is between sunrise and sunset and between sunset and sunrise. The color code is shown in the top-right panel. The plain and hatched bar plots represent the cases without and with aerosol perturbation, respectively. Note that the vertical axis is different for $\overline{SWUP}$ and $\overline{LWUP}$.

The insights gained from this study would help to address the efficacy of MCB as a potential climate intervention strategy to reduce global mean temperature. Our results show that there is a substantial decrease in $f_c$ in the warmer world (SSP1 and SSP3 scenarios). This is also supported by the long-term observation trend (Norris et al., 2016). In particular, in our simulations we find that the cases with a drier FT appear to be more susceptible to warming-induced $f_c$ reduction. All of this points to a significant reduction in the potential efficacy of MCB as climate change progresses. Thus, from an operational perspective, it would be more effective if MCB is deployed sooner rather than later, thus buying additional time for decarbonization technologies to become viable.

## 5 Summary and Outlook

In this study, we explored aerosol-cloud interactions (ACI) in a warmer climate in marine low-clouds using Lagrangian (domain advecting with mean wind) large-eddy simulations (LES) coupled to two-moment, bin-emulating bulk microphysics models.



We used the well-explored composite reference case from Sandu and Stevens (2011) to set up the stratocumulus-to-cumulus
transition (SCT) simulations. In this case, precipitation plays a prominent role in the transition (precipitation-mediated tran-
sition) (Yamaguchi et al., 2017). Since not all SCTs are driven by precipitation, we created another case with reduced FT
humidity. The humidity was reduced to 27% of the value in the composite reference case in Sandu and Stevens (2011). This
resulted in a case where the onset of cumulus activity was driven by the classical entrainment-deepening of the marine bound-
ary layer (entrainment-driven transition). The equivalent cases for warmer conditions at the end-of-the-century were created
following the methodology laid out in Bretherton and Blossey (2014). The large-scale forcings for this setup were obtained
from the latest coupled model intercomparison project (CMIP6) data. We considered two shared socioeconomic pathways
(SSP) from CMIP6: (i) SSP1-2.6: this is the best-case scenario with a sustainable future with clean up, and (ii) SSP3-7.0:
fossil-fueled growth due to regional rivalry.

Our results showed that the effect of climate change on the cloud radiative effect (CRE) dominates over the effect of aerosol
perturbation prior to the onset of precipitation in both precipitation-mediated and entrainment-driven cases. In particular, the
impact of climate change is much stronger in the entrainment-driven case compared to the precipitation-mediated case. This is
because of the higher LWP in the precipitation-mediated case, which shields the cloud fraction ($f_c$) from erosion against the
weakened radiative cooling and warming effects associated with climate change. Interestingly, the impact of climate change is
muted once the cloud deck is broken up into open-cellular cumulus clouds.

We also explored the impact of climate change on the onset of cumulus activity, as quantified using the variance in cloud
base height. In the moist FT case (precipitation-mediated), the effects of climate change resulted in an earlier onset of cumulus
activity, which advanced the onset of precipitation-induced cloud breakup in a strongly warmer climate (SSP3). In the dry
FT case (entrainment-driven), the onset of cumulus activity appeared immune to the effects of climate change. Our analysis
indicated that the weaker radiative cooling associated with higher humidity in the FT makes the boundary layer less coupled
in the precipitation-mediated case. Further weakening of radiative cooling associated with climate change enhances the decou-
pling of the boundary layer, resulting in an earlier onset of cumulus activity. A substantially higher cooling rate at cloud-top
in the entrainment-driven case (due to the lower humidity in the FT) maintained a well-mixed boundary layer in all cases until
the onset of cumulus activity on day 3. In this case, a similarity in the onset time of cumulus clouds does not translate into a
similarity in the onset time of cloud breakup. In a strongly warmer climate (SSP3), the onset of cloud breakup is substantially
delayed due to its lower LWP and weaker precipitation intensity. In both moist and dry FT cases, aerosol injection delayed the
breakup of the stratocumulus layer through precipitation-suppression, thus substantially enhancing the magnitude of CRE in
all scenarios.

The central focus of our study was to assess the impact of climate change on ACI in marine low-clouds. Our analysis showed
that the $N_d$ and LWP components of the cloud albedo changes due to aerosol perturbation are similar under the influence of
climate change. Thus, in non-precipitating clouds, to leading order, the changes in CRE due to aerosol perturbation (dCRE)
normalized by $f_c$ ($\frac{\mathrm{dCRE}}{f_c}$) are constant under the influence of climate change. However, the albedo contributions associated with
precipitation-suppression are non-linearly coupled to $f_c$ changes and cloud albedo, and are not immune to climate change.
Despite this similarity in the ACI in a warmer climate in non-precipitating clouds, the efficacy of marine cloud brightening



(MCB) reduces substantially towards the end-of-the-century due to the reduction in cloud fraction associated with the effects
of climate change. Moreover, the dCRE in precipitating clouds also decreases due to the effects of climate change. Thus, if
considered, a practical implementation of MCB would be more efficient sooner rather than later.

The simulations carried out here are based on an idealized SCT case study built from a 2-year (summer-time) composite
trajectory that does not account for all the real-world complexity and variations (e.g., changes in wind speed). Additionally, the
insights from the current study and similar LES studies are only relevant in assessing ACI in the initial phase of MCB. Sustained
MCB will result in significant changes in large-scale climatology and background aerosol concentration. To overcome these
challenges, future studies should focus on an ensemble of trajectories based on realistic conditions in the present and warmer
world to assess the impact of climate change on the efficacy of MCB, changes in ACI, and SCT. These studies would aid
in assessing the generality of the key hypotheses from the current study - (i) changes in CRE due to aerosol perturbation
normalized by $f_c$ ($\frac{\text{dCRE}}{f_c}$) in non-precipitating clouds are invariant under climate change, and (ii) the efficacy of MCB decreases
in a warmer climate due to the reduction in cloud fraction.

*Code and data availability.* The simulations were carried out using SAM (https://wiki.harvard.edu/confluence/display/climatemodeling/SAM).
The data for reproducing the plots will be uploaded after acceptance.

*Author contributions.* PP and GF designed the research with inputs from TM. PP carried out the simulations and analysis. PP, TM, FH, and
GF discussed the results. PP wrote the manuscript with input from TM, FH and GF.

*Competing interests.* At least one of the (co-)authors is a member of the editorial board of *Atmospheric Chemistry and Physics*. The authors
have no other competing interests to declare.

*Acknowledgements.* This research was supported by the U.S. Department of Commerce, Earth's Radiation Budget grant, NOAA CPO Climate and CI no. 03-01-07-001. FH appreciates support from the Emmy Noether program of the German Research Foundation (DFG) under
grant HO 6588/1-1. Prof. Marat Khairoutdinov graciously provided the SAM model. PP thanks Dr. Jan Kazil and Dr. Isabel McCoy for their
inputs in setting up the warmer world simulations.





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
