# Peer review of "Aerosol-Cloud Interactions in Marine Low-Clouds in a Warmer Climate"

_EGUsphere, 2025_

## Author Comment (AC1)

**Response to Reviews of "Aerosol-Cloud Interactions in Marine Low-Clouds in a Warmer Climate"** - Prabhakaran *et al.* 2025

**Reviewer 1:**

Summary:
The study investigates how climate change and aerosol perturbations influence the stratocumulus-to-cumulus transition over the North-East Pacific. Using Lagrangian large-eddy simulations along idealized trajectories, the authors examine two distinct free-tropospheric humidity conditions, moist and dry, which aims to control the transition mechanism (precipitation driven vs entrainment driven). They analyze two climate scenarios (SSP1-2.6 and SSP3-7.0) from CMIP6 to capture a range of warming and aerosol changes. Results show that in non-precipitating stratocumulus, the CRE is more sensitive to climate change than to aerosol perturbation. However, after the cloud deck breaks up, aerosol effects, mainly through precipitation suppression, become more influential. The study also finds that aerosol–cloud interactions such as the Twomey effect and LWP adjustments are mostly robust to climate change unless cloud fraction is significantly altered.

The topic addressed in the manuscript is important and interesting, with relatively few publications on the subject to date. I therefore think it is valuable for this manuscript to be published. Given the complexity of the problem, which involves considering both changes in meteorology and aerosol effects, I think the authors have done a nice analysis. I do have a few points that I believe should be clarified before publication:

We thank the reviewer for the constructive feedback. In our response below, our responses are in blue and the reviewer comments are in black.

General:

1. The domain size of the simulations is 48 km. Given that the SCT involves boundary layer deepening and the development of wide convective cells, I am concerned that a 48 km domain may be too small to adequately capture the full extent of boundary layer circulations. If the characteristic cell size exceeds the domain size, can the simulation reliably resolve the SCT dynamics?
The characteristic cell size plays an important role in the SCT dynamics involving precipitation. This was explored in great detail in Yamaguchi et al 2017. They showed that the onset of precipitation, associated feedback, and the subsequent cloud breakup is sensitive to the domain size. Yamaguchi et al 2017 used the Sandu & Stevens 2011 composite case for their study and concluded that a domain size of 24 km was sufficient to capture the effects of precipitation and associated feedback in SCT. The moist case in the current study is based on Sandu & Stevens et al 2017. Therefore, we are of the opinion that a domain size of 48 km is adequate to capture the effects of convective cells for the moist cases considered in our study. Additionally, we also ran a 128 km domain simulation for the Sandu & Stevens 2011 case in Prabhakaran et al 2024. Our results (48 km domain) are similar to the results in Prabhakaran et al 2024.

For the dry cases, since precipitation plays a secondary role, the importance of convective cell sizes in SCT is not clear and requires further investigation, which is beyond the scope of the current study. In the revised draft, we have included a brief discussion in the Methodology section to address this comment (Lines 87-92).

2. The simulations are based on the composite trajectories from Sandu and Stevens (2011). However, if aerosol–cloud interactions are nonlinear with respect to environmental conditions, the use of a composite trajectory may not reliably capture the true impact of climate change on cloud properties. Could the authors comment on the extent to which this approach might mask important variability or lead to biased conclusions?
We agree with this point and have already acknowledged the same in the Summary and Outlook section (Last paragraph). To account for the variability in meteorological conditions we have recommended that the future studies use an ensemble approach. Until these or similar studies are carried out, it

would not be possible to quantify the extent of the biases from the use of a composite trajectory. We intend to address these issues in a future study.

Since the main point is already acknowledged in the manuscript, we do not make any changes to address this comment.

3. To create two types of SCT breakup (entrainment-driven and precipitation-driven), the authors reduce the humidity to 27% of the reference value, which leads to an entrainment-driven transition. However, by manipulating the humidity in this way, aren't the authors effectively prescribing specific climate change scenarios characterized by a drier free troposphere?

See below.

4. I am unclear about the need to lower the free-tropospheric humidity in order to trigger an entrainment-driven breakup. As I understand it, the trajectories from Sandu and Stevens (2011) are intended to represent typical SCT cases that include entrainment-driven breakup. If that is the case, wouldn't increasing the humidity be the approach needed to shift the system toward a precipitation-driven breakup instead? Clarification on this point would be helpful.

Response to comments 3 & 4: The composite trajectory in Sandu & Stevens 2011 averages over all possible SCTs in the 2-year period from 2006-2007 (summertime - JJA). Sandu & Stevens 2011 and a subsequent study by Bretherton & Blossey 2014 did not find a significant role for precipitation in their simulations. This conclusion was an outcome of fixed cloud droplet concentration in their microphysics scheme and smaller domain size (Yamaguchi et al 2017). Our interactive microphysics model allows local scavenging of aerosol that increases the chance of precipitation, which further accelerates precipitation. A fixed or lower cloud droplet concentration does have the same effect. Several earlier studies with interactive microphysics (Yamaguchi et al 2017, Zhou et al 2017, Prabhakaran et al 2024) have shown that cloud breakup in Sandu & Stevens 2011 reference case is associated with precipitation, resulting in open-cellular clouds.

The FT in the NEP region is quite dry and SCT is very often driven by entrainment deepening (Bretherton et al 2019, Eastman et al 2021). To make our study more representative and to reduce bias, we created the dry case (reduced FT humidity) to delay the onset of precipitation, thus allowing enough time for the cumulus clouds to develop.

In our study, climate change is represented by the changes in $CO_2$ concentration, SST, and FT humidity. Note that the change in FT humidity due to climate change is related to the change in SST as the FT relative humidity is the same in PD and end-of-century conditions (Bretherton & Blossey 2014). Reduction in FT humidity alone does not characterize climate change. It merely represents a different realization in a given climate scenario.

We have modified the following parts of the Methodology section: Lines 75-77 and 81-86 in the revised manuscript.

5. The paragraph starting at line 157 and the following one are quite difficult to follow. More generally, the results are highly descriptive throughout the manuscript. In my view, the level of detail is overly comprehensive, which makes it challenging to follow. It may be helpful allow the figures to also speak for themselves.

We thank the reviewer for pointing this out. We have rewritten parts of the results section (Section 3) to address this comment. We have pulled back on some of the details to improve the readability. The changes are present throughout Section 3. Please check the tracked-changes file ("diff.pdf") for changes.

6. I think that a significant portion of the current discussion section would be more appropriately placed in the results section. Several parts read more like continued presentation of results rather than higher-level synthesis.

We thank the reviewer for this thoughtful comment. We agree that portions of the Discussion section contained detailed descriptions that might have appeared like an extension of the Results section. However, we found that moving them to Results section affects the overall flow of the arguments and interpretation. Instead, we have rewritten parts of the discussion section (Section 4.1 in particular) so that we focus more on the broader interpretation and synthesis.

In the revised draft, most of the changes are between lines 282 and 296. Please see the attached "diff.pdf" file for all the changes in Sec. 4.

Specific comments:

1. Figure 4: why are the dashed lines in subplot 1?
That was an oversight on our part. We have replaced the dashed lines with solid lines in Figure 4.

2. Line 219-224. This part is somewhat confusing, as it begins with a focus on the entrainment-driven transition (as indicated by the subsection title), but then shifts to emphasizing the role of precipitation.
In the entrainment-driven case, the transition is not initiated or driven by precipitation. Precipitation is initiated several hours (8-10 h) after the onset of cumulus activity. We emphasize the role of precipitation because of its importance to CRE. We have clarified this in Lines 222-224 in the revised draft.

3. Figure 7: You could add "Moist" and "Dry" labels above the left and right columns.
We have updated the figure and have added "Moist" and "Dry" labels.

4. Equation 4: Could you clarify the role of this equation in your analysis? It's not clear how it is used in the manuscript.
We do not use Eq. 4. We proposed Eq. 4 as a potential way to include the contribution of ACI to the changes in CRE in non-precipitating marine low-clouds. In particular, it highlights the invariance of $\frac{\partial A_c}{\partial N_d}$ under climate change. We are not using it to quantify ACI. We leave it to future studies to assess the validity of our hypothesis using this equation. We have updated the manuscript to address this comment. See lines 351-354.

5. Line 348: "meteorological variables or cloud controlling factors" – Aren't meteorological variables essentially the same as cloud-controlling factors? If not, could you clarify the distinction? If they are the same, using "or" may be misleading.
We have modified the text. We use meteorological factors. See lines 348 in the revised text.

6. Line 451: You mention here " the albedo contributions associated with precipitation-suppression are non-linearly coupled to fc changes and cloud albedo" (this is also stated earlier in the manuscript). It is unclear to me what exactly is meant by "non-linearly coupled" in this context. Could you clarify the nature of this coupling?
We meant to say that with precipitation-suppression both $f_c$ and $A_c$ are affected. This results in a multiplicative effect on cloud radiative properties. This is evident from Eq. 1 as well. To avoid confusion, we have removed all references to "non-linear coupling". See lines 185, 352-354, 449-450 in the revised manuscript.

7. Isn't the mathematical form in line 451 is redundant? Mathematically it reduces to dCRE. Perhaps a comma is missing after fc.
We agree that this was confusing and have modified the text by removing the mathematical form. See lines 448-450 in the revised manuscript.

**Reviewer 2:**

Summary:

The paper "Aerosol-Cloud Interactions in Marine Low-Clouds in a Warmer Climate" by Prabhakaran et al. explores the relationship between aerosols, clouds, and climate warming on stratocumulus-to-cumulus transition using large eddy simulations model simulations. The paper is suitable for publication in the journal Atmospheric Chemistry and Physics. The paper is well written and the research question is novel and so tt meets the journal's requirements in terms of scientific scope, novelty, methodological rigor, and formal structure. I can recommend publishing the paper once the following minor comments have been addressed:

We thank the reviewer for the positive evaluation of our paper and for the constructive feedback. In our response below, the reviewer comments are in black and our responses are in blue.

Line 99. The use of CMIP6 data is ambiguous, what does "the global mean outputs from CMIP6" mean? Is it the CMIP6 ensemble mean for the region?
We used the global average from the CMIP6 ensemble. We do not consider regional variations. In the revised manuscript, we have stated this explicitly in the methodology section. Lines 106-109.

Line 100. How does the LES take into account the CO2 concentration?
CO2 concentration is provided as an input to the radiative transfer model RRTMG. The resulting change in radiative heating profiles is used by the LES model to simulate the dynamics.

Line 124. It would be useful to the reader to mention how much earlier the cloud breakup occurs.
We have stated this in the revised manuscript (by approximately 10-12 hours). See lines 133-134 in the revised manuscript.

Lines 188-189. "Additionally, on day 2 there is suppression of weak precipitation from the addition of aerosol in SSP1 (Fig. 1e)". This should be Figure 1f?
Thank you for pointing this out. We have fixed this. See line 171 in the revised manuscript.

Line 326: What is meant by better in "the better collapse in the LWP component"
By "better collapse" we meant reduced scatter. We have replaced "better" with "improved". See lines 324-326 in the revised manuscript.

Since the study is based on a idealized case, it is not clear if it's results can be extrapolated to different types of meteorological conditions and this is also acknowledged in the conclusions. However, it would be comment briefly if this approach can be used to estimate the efficacy of artificial marine cloud brightening since the method of spraying sea water does not only affect the amount of aerosols, but the evaporation of sea water would also have implications on the boundary layer moisture and temperature profile.
While it is true that the results from the current idealized study cannot be directly extrapolated to different meteorological conditions, the qualitative insights from this study will be useful. For instance, we have shown that the efficacy of MCB will decrease in a warmer climate. The extent to which this happens will depend on the background meteorological and aerosol conditions.
The impact of evaporation of sea water on MCB is a relatively under explored topic. Jenkins and Forster 2013 (DOI: 10.1002/asl2.434) have argued that evaporation of sea water would reduce the efficiency of MCB. Since Jenkins and Forster 2013 only focused on a single case study, it is not clear if this study would be relevant under all meteorological conditions. More studies are required to explore this problem. We have briefly mentioned this in the summary and outlook section. See lines 458-460 in the revised manuscript.

---

## Author Response (AR2)

We thank the editor for the thoughtful comments. Our response to these comments are marked in bold below.

1) I agree with reviewer #1 that a lot of section 4 is a continuation of the results rather than a discussion extending beyond the findings of your analysis. This has not been sufficiently addressed in your revision. This holds especially for sections 4.1, 4.2, and 4.4. This does require a more thorough reorganisation of the text as done so far.

**We have revised the text to address this concern. We have moved major portions of the previous discussion section to the results section and created a new section (Sec. 4). The revised discussion section is substantially shorter. Please see diff_ed.pdf for tracked-changes.**

2) Please add your aerosol perturbation flux in L104 in the methods section for clarity and reproducibility.

**We have added that in L 104.**

3) You may wish to extend your discussion section on the realisms of your climate change scenarios and potential limitations in context to potential stabilising effects between the sub-tropics and tropics that are unaccounted for in your climate change setup.

**We have included a brief discussion about the lack of coupling between the tropics and sub-tropics in our study. Lines 456-459 in the revised draft. We also point out that there are a few studies (e.g. Chun et al 2025) that attempt to model the coupling between the tropics and sub-tropics using the WTG approximation. However, there is quite a bit of uncertainty regarding the accuracy of such methods/approximations. For instance, the presence of secondary circulations (e.g. shallow meridional circulations Zhang et al 2004 Jou. of Climate) may introduce additional time scales into the problem which are not considered in the WTG implementation. A more detailed analysis will be required to address these issues.**

**In our Summary and Outlook section, we present a brief picture of the above-mentioned discussion. Lines 456-459 in the revised draft.**

Additional private note (visible to authors and reviewers only): Editorial comment: I personally find your colour choices misleading. "red" for present-day (less warm) conditions? I leave it at your discretion to change or keep this.

**We are comfortable with this color code. Black - SSP3 is the worst case scenario with very high emission rate and green - SSP1 is the most optimistic scenario with substantial clean up. Red - PD, present day is already in dangerous territory due to high emission rates.**